# Effects of ASL Rhyme and Rhythm on Deaf Children’s Engagement Behavior and Accuracy in Recitation: Evidence from a Single Case Design

**DOI:** 10.3390/children7120256

**Published:** 2020-11-26

**Authors:** Leala Holcomb, Kimberly Wolbers

**Affiliations:** Jane and David Bailey Education Complex, University of Tennessee, Knoxville, TN 37996, USA; kwolbers@utk.edu

**Keywords:** deaf, preschool, rhyme, rhythm, recitation, engagement, language processing, phonological awareness, sign language, language deprivation, language impairment, single case study, alternating treatments

## Abstract

Early language acquisition is critical for lifelong success in language, literacy, and academic studies. There is much to explore about the specific techniques used to foster deaf children’s language development. The use of rhyme and rhythm in American Sign Language (ASL) remains understudied. This single-subject study compared the effects of rhyming and non-rhyming ASL stories on the engagement behavior and accuracy in recitation of five deaf children between three and six years old in an ASL/English bilingual early childhood classroom. With the application of alternating treatment design with initial baseline, it is the first experimental research of its kind on ASL rhyme and rhythm. Baseline data revealed the lack of rhyme awareness in children and informed the decision to provide intervention as a condition to examine the effects of explicit handshape rhyme awareness instruction on increasing engagement behavior and accuracy in recitation. There were four phases in this study: baseline, handshape rhyme awareness intervention, alternating treatments, and preference. Visual analysis and total mean and mean difference procedures were employed to analyze results. The findings indicate that recitation skills in young deaf children can be supported through interventions utilizing ASL rhyme and rhythm supplemented with ASL phonological awareness activities. A potential case of sign language impairment was identified in a native signer, creating a new line of inquiry in using ASL rhyme, rhythm, and phonological awareness to detect atypical language patterns.

## 1. Introduction

The difference between hearing and deaf toddlers’ early language access and experience is stark. From birth, hearing children access the playful phonological patterns of spoken language on a minute-by-minute basis [1]. Hearing children benefit greatly from language activities that incorporate rhythmic and rhyming spoken language (e.g., nursery rhymes or Dr. Seuss), especially if paired with spoken phonological awareness tasks [2,3,4]. What has become ubiquitous among households, daycare, and early childhood education programs for hearing children is largely absent or inaccessible to deaf children. Deaf children can neither fully nor easily access sound, limiting their ability to fully benefit from spoken rhyme and rhythm. As we will see, early childhood education for deaf children is only just beginning to incorporate American Sign Language (ASL) rhyme and rhythm in the classroom. There are different reasons for this gap between deaf and hearing children; one of these reasons is the lack of comparable research as to the benefit of ASL rhyme and rhythm for deaf children’s phonological awareness and language development. This study adds to a growing body of research aimed at addressing this gap.

While there are historical documents on language play in American Sign Language (ASL)—such as on rhyme, rhythm, and poetry in the deaf community—this practice has been excluded from most early childhood classrooms that serve deaf children [5]. The absence of early language approaches that utilize ASL can be explained by the long history of the stronghold of oralism (speaking-only) in Deaf education [6]. The exclusion of deaf teachers meant that their unique cultural and linguistic approaches were also precluded from deaf children’s language experiences. It was not until the 1970s that ASL began to be embraced in a few deaf schools. A contributing factor to this change was the recognition of ASL as a true, sophisticated language by linguists [7]. Since then, there is accumulating evidence that deaf children immersed in a signing environment during their early years can meet language milestones, develop robust literacy skills, and excel in academic studies [8,9,10]. Yet negative biases towards the use of ASL with deaf toddlers have not wavered, especially during the early childhood period [11]. Although tremendous potential is evident, approaches that maximize deaf children’s visual access to language have still not fully infiltrated the educational system.

In small pockets of deaf education, some doors have opened for deaf educators and leaders to enter the field and devise methods of language instruction that are culturally and linguistically responsive to deaf ways of learning [12]. ASL rhyme and rhythm were among the approaches that materialized in the ASL/English bilingual classroom [13]. Deaf educators and researchers have recently begun to come together to create new ASL rhyming and rhythmic resources in response to the demands from teachers and families (e.g., ASL Mother Goose Program and Hands Land). We ask whether parallel forms of language intervention using ASL rhyme and rhythm can support engagement and recitation skills in deaf children. This single-subject study compared the effects of rhyming and non-rhyming ASL stories on the engagement behavior and accuracy in recitation of five deaf children between three and six years old in an ASL/English bilingual early childhood classroom.

### 1.1. Literature Review

Chukovsky [14] documented hearing children’s fascination with language and their tendency to make up rhyming words. His own four-year-old child was found spontaneously screaming, “I’m a big, big rider! You’re smaller than a spider!” Hearing children are regularly exposed to rhyme and rhythm packed with patterns and repetitions in sound that propel language development. Rhymes are the same sounds produced across different words such as in this verse: “the fat bat sat on the cat’s hat”. Rhythms are the patterned stressed and unstressed beats that sounds make. The developing human brain is biologically captivated by these patterns and fires neurons to build robust connections [15]. The consistent and abundant input of rhyme and rhythm on a daily basis through incidental exposure in the environment (e.g., restaurants, grocery stores, car rides, television) and explicit instruction (e.g., parentese, daycare centers, early childhood classrooms, interventions) make a significant impact on hearing children’s development.

There are sequential steps in language acquisition that follow a timetable. As infants begin to acquire language, they attempt to produce their first sounds, words, and then songs by imitating their caregivers’ language input. Rhyme, rhythm, and parentese (baby talk) are among the critical early language approaches utilized for healthy brain development [16]. With rich language input, toddlers’ language grows exponentially between two and five years old, enabling them to turn their attention to language itself through the development of phonological awareness. Phonological awareness is the knowledge and understanding of how sound structures work [17]. Adams [18] presented a hierarchical trajectory of phonological awareness development. Following the sequence of development, hearing children first develop sensitivity to and awareness of sounds by recognizing that they can be put together to form words. With this awareness, they begin to enjoy playing with language and reciting nursery rhymes. Next, they learn that some words share similar sounds while others do not. They recognize words that rhyme such as ‘cat,’ ‘fat,’ and ‘bat’. Then, they are able to put together individual sounds to produce a word. They can say “bat” after being given the individual sounds /b/a/t/. Inversely, they can break down a word into individual sounds by saying /b/a/t/ when given the word ‘bat’. Finally, they can manipulate sounds in words through deleting or substituting. If asked to say ‘fear’ without the /f/, they would be able to say ‘ear.’ If asked to replace individual sounds to create new rhymes, they would be able to say ‘bat,’ and ‘sat,’ or ‘sit’ and ‘sip.’ With these language opportunities, children often enjoy playing with sounds by singing out aloud by themselves or with each other—dee, dee, bee, bee, tee, tee, rub, rub, dub, dub. With a strong foundation in language, hearing children are primed to navigate the challenges and complexity of literacy development.

A meta-analysis by the National Early Reading Panel [19] concluded that access to lexical store, phonological memory, and phonological awareness during early years is a good predictor for literacy success. This finding signifies the importance of using approaches that maximize young children’s language input, and that includes rhyme, rhythm, and phonological awareness [2,4,20,21,22,23]. Patscheke et al. [22] investigated the impact that music and phonological training had on phonological awareness in four- to six-year-old children of immigrant families. Thirty-nine preschoolers were randomly assigned to three groups and received a twenty-minute intervention thrice a week. One group received music training along with phonological training. One group received phonological training only. One group served as the control and received sports training. Children in the music group and phonological group significantly increased their scores on the phonological awareness test. However, the effect size of the music group was much larger compared to the phonological group. This finding supports the hypothesis that combining music with phonological awareness activities increases language outcomes. Further expanding on evidence, Dunst and Gorman’s meta-analysis [24] also found that all children, regardless of differences in disability, age and gender, had positive outcomes in their language and literacy development after exposure to rhyme and rhythm.

The widespread use of music in every culture makes it common for hearing children to recognize rhymes and enjoy reciting songs without specialized interventions. Hearing children with language disabilities, however, may struggle with phonological awareness tasks and need more explicit instruction. When hearing children show difficulty with imitating or reciting common nursery rhymes, professionals raise red flags for dyslexia or language impairment [25]. Multiple interventions that utilize rhyme and rhythm exist for the purpose of helping develop phonological awareness in hearing children [4]. Similarly, activities to increase engagement and recitation skills are typically a part of interventions in early childhood education.

### 1.2. Engagement and Recitation

The construct of engagement is defined as having cognitive, affective, and social components [26]. First, there is the cognitive task of having heightened alertness and focused attention. Second, there is the affective behavior of having positive attitudes towards the language and the willingness to use it. Third, there is the social aspect of initiating interactions using language. Rhyme and rhythm are known to increase attention, motivation, imitation, and interaction in hearing children, especially for those with disabilities [27]. A meta-analysis found that there were large effects on attention and motivation from music therapy for hearing individuals with developmental or behavioral disabilities [28].

A single-subject mixed method study conducted by Vaiouli et al. [27] looked into the effectiveness of a music intervention in engaging three hearing kindergarteners with autism. Each music therapy session included a welcome song, a child-led part, an adult-led part, and a goodbye song. Actions that counted as engagement behaviors were the child’s instances of focusing on the adult’s face, shifting gazes, showing awareness and positive affect with smiles or nods, exchanging looks between the object and the adult, and pointing at or showing objects. Significant gains in engagement behaviors were seen in all three children. Similarly, Perry [29] conducted a qualitative study observing 10 hearing students with multiple disabilities during music therapy sessions. Music activities led to increased attention, turn taking, and expressive language. Some children exhibited behaviors of interest and attention during musical activities that they rarely demonstrate in routine activities. Not only do rhyme and rhythm promote engagement in hearing children, but they also support recitation skills.

Recitation is defined as repeating language aloud from memory. Research shows that being exposed to rhyme and rhythm enriches the ability to recite [30,31,32,33]. Calvert and Billingsley [30] looked at whether hearing children were able to recite songs without understanding the meaning of words. They showed an incomprehensible song in French and a comprehensible song in English on television to 48 English-speaking preschoolers and asked them to recite the song verbatim. Children were able to successfully recite the incomprehensible French version of the song in the same way as they recited the comprehensible English song. Children’s ability to process and recite words did not depend on their knowledge and understanding of these words, indicating the effects of rhyme and rhythm on recitation skills.

Sheingold and Foundas [34] examined the impact of rhyming and non-rhyming versions of stories on 24 five- and six-year-old hearing children’s ability to accurately recall the details and provide the correct picture sequence of stories. More specifically, the researchers wanted to know if the presence of rhyme would impact memory. Both versions (rhyming and non-rhyming) of each story had the same information but with rhyming words removed in the non-rhyming version, and they were counterbalanced in their administration. After the story was told, the child was asked five questions about the content of the story and arranged the picture cards in the correct sequential order. The researchers found that hearing children did better with the rhyming version. More children also chose the rhyming version as their favorite over the non-rhyming version, supporting social validity.

Johnson and Hayes [33] examined the effects of rhyme on 64 preschoolers’ recitation of stories by comparing their performance in reciting rhyming and non-rhyming versions of a short story. Both versions were similar in content but had different order of the lines in the stanza to remove the rhyming aspect in the non-rhyming version. Their dependent variable measures were the numbers of story words correctly recited and the number of story words recited in the correct presentation order. The results of two-factor analysis of variance yielded information that the rhyming version increased verbatim recitation in correct sequential order. However, children also did adequately well with paraphrasing the non-rhyming version. Preschoolers were able to perceive and process the phonological patterns, leading to heightened ability to recite rhyming stories. Yet non-rhyming stories still served their purpose in facilitating comprehension and paraphrasing. This study provides evidence of different kinds of language processing and their distinct benefits when it comes to recitation and memory, which serve language development.

Read et al. [35] conducted a group experimental study to see if rhyming words in shared storybook reading helped hearing children retain more words. They split 24 children aged two to four years old into two groups and had parents read either a rhyming or non-rhyming version of the same animal story to their child individually. The results showed that children were able to retain more words in the rhyming condition, supporting the hypothesis that exposure to rhyme boosts word retention and vocabulary development. These findings illustrate the ways rhyme, rhythm, phonological awareness, engagement, and recitation benefit hearing children.

### 1.3. Deaf Learners

Unlike the literature on the role of spoken rhyme, rhythm, and phonological awareness for hearing children, little is known about the benefits of ASL rhyme, rhythm, and phonological awareness for deaf children. While research shows that ASL does have its own ways of generating rhyme and rhythm through phonological play, syllables, and movements, there is still much to learn about their relevance to language acquisition and emergent literacy. A review of the literature provides a glimpse into the potential of ASL rhyme, rhythm, and phonological awareness sharing similar functions for deaf children as spoken rhyme and rhythm does for hearing children.

Valli [36], a deaf person known for his ASL poetry renditions, conducted his dissertation examining the role of eye gaze, body shift, head shift, handshapes, and movement in creating rhyme and rhythm in ASL poetry. He explored strategies used by teachers in teaching ASL poetry to deaf students. The teachers incorporated phonological patterns in instruction that stressed repeated patterns in handshape, location, and movement. Valli saw that this instruction helped deaf children understand, memorize, and create ASL poetry through visual rhymes. Valli’s work brought new knowledge to the field regarding the application of ASL linguistics to ASL poetry within the context of deaf education. Investigations into signed language rhyme and rhythm were also conducted overseas in France.

Blondel and Miller studied the uses of nursery rhymes in French Sign Language (LSF). They said, “… nursery rhymes exist in sign languages. They are part of language games, along with tongue-twisters, lullabies, riddles, and so on. As far as we know, they are created by deaf adults for children” [37] (p. 29). Phonological parameters required for creating rhymes were identified: handshape, location, movement, and non-manual markers. In LSF, just like ASL, rhythm can be formed by maintaining the flow and manipulating the transitions of signs to make the initial parameter (i.e., handshape, location, movement) of the sign match the previous or subsequent signs. Furthermore, syllables are found in the movements and holds in signed words. The researchers marked the benefits of developing original nursery rhymes in sign language rather than translating spoken nursery rhymes, which had linguistic and phonological limitations.

In Canada, original ASL rhyme and rhythm were developed for a parent-infant program called the ASL Parent-Child Mother Goose Program [38]. In interviews, hearing parents shared excitement in witnessing their deaf toddlers responding positively to ASL rhyme and rhythm by babbling along with their arms and hands, producing happy facial expressions, and laughing. These deaf toddlers’ behaviors did not depart from what was expected from hearing toddlers when exposed to spoken rhyme and rhythm. Because deaf toddlers demonstrate favorable responses while viewing ASL rhyme and rhythm, some early childhood educators are beginning to see the benefits of incorporating these practices into their instruction.

Crume observed and interviewed teachers of the deaf in a school that was converted into an ASL/English bilingual program in the 1990s. It was discovered that rhyming and rhythmic activities were recently introduced to their early childhood program. Crume defined the sign rhythm activity as “ASL stories incorporating signs with repetitive movements” [13] (p. 105). A teacher said that they were experimenting with ASL rhyme and rhythm with preschoolers by repeating signs with the same handshape in rhythmic movements. Crume expanded on the definition of this practice in his observation, “In sign rhythm activities, teachers incorporate the repetitive use of signs together with clapping or patting on knees. The sign rhythm activities allowed deaf students to learn specific handshapes in signs in a pattern that made learning fun. This provided the deaf students a similar benefit that hearing preschool children enjoy when they incorporate movement and gesture in songs” [13] (p. 99). Teachers remarked on the effectiveness of rhyme and rhythm activities in increasing engagement and motivation in students with limited language.

Researchers identified a predictable progression in phonological skills over the course of natural acquisition of sign language in young children [39,40,41,42]. Young children are able to produce signed words in correct locations before they learn to use correct handshapes and movements. Di Perri’s experimental study [43] on phonological awareness in 29 deaf children between four and eight years old further reinforces these findings. Deaf children were asked to engage in the tasks of identification, categorization, differentiation, blending, segmentation, and substitution for each phoneme—handshape, location, and movement. Location was found to be the easiest phoneme for deaf children to manipulate, followed by handshape, and then movement. Categorization was the easiest phonological awareness task, followed by identification, and then differentiation. More tasks such as blending, segmenting, and substituting parameters in signed words were also examined. It was discovered that segmenting a signed word into parameters was the easiest, followed by blending, and then substituting. No significant difference could be found in performance across tasks and ages. The hierarchy of ASL phonological awareness parallels English phonological awareness, with the task of identifying being the easiest and substituting being the hardest [18], indicating similarities in language development processes.

Andrews and Baker [44] argued that ASL rhyme and rhythm should be used in early childhood programs to support healthy language acquisition. They laid down a framework in which ASL rhyme and rhythm can promote ASL phonological awareness, ASL vocabulary development, and emergent literacy. Among the prominent advantages of ASL rhyme and rhythm are early communication, executive functions, language, early literacy, and metalinguistic awareness. The authors asserted that ASL rhyme and rhythm are a valuable tool in making language experiences positive and fun for young deaf children and their families alike. Although it may be intuitive to say that ASL rhyme, rhythm, and phonological awareness benefit deaf children, findings from cognitive neuroscience further add to the strength of the argument.

Petitto [15,45,46,47] has carried out some groundbreaking work in neuroscience research on identifying the functions of ASL in the developing brain. Her twenty years of research have extensively expanded our knowledge of literacy development in relation to brain mechanisms and phonological awareness in deaf children. Whether the modality is spoken or signed, the brain responds in the same way in its search for phonological patterning. Petitto and her colleagues said, “the crucial link for early reading success is not between print and sound, but between print and the abstract level of language organization that we call phonology - signed or spoken…” [15] (p. 367). Similar conclusions were made by McQuarrie and Abbott, who have done substantial work in exploring the role of ASL phonological awareness in literacy development in deaf children. “Having a strong phonological foundation in any language may be more important than the modality itself” [48] (p. 96). In terms of the relationship between phonological awareness in English and literacy skills, a meta-analysis conducted by Mayberry et al. [49] found that English phonological coding and phonological awareness skills accounted only for 11% of the variance in reading proficiency in deaf participants. Further affirming this phenomenon, a recent experimental study tested deaf children’s English vocabulary scores after viewing ASL-English bilingual stories that utilized ASL rhymes, ASL-English bilingual stories without any rhymes, and ASL-English stories that had English rhymes. Results showed that deaf children demonstrated the greatest gains in English vocabulary scores with the ASL rhyming story over the other two conditions [50]. These findings suggest that ASL rhyme, rhythm, and phonological awareness may have a bigger role in language and literacy development than previously realized.

While there is no clear explanation yet on exactly how deaf children transfer their ASL knowledge to the acquisition of English, what can be derived from the current literature is that deaf children can become successful readers through myriad avenues either with or without spoken phonological awareness [8] and that ASL phonological awareness may have a positive influence on both ASL and English literacy [48,50,51]. How the use of two languages—ASL and English—are processed and mediated in the brain during the acquisition and development phase in bilingual deaf children remain understudied. Similarly, the impact of ASL rhyme, rhythm and phonological awareness on deaf children’s language has not been thoroughly investigated.

### 1.4. Sign Language Impairments and Language Deprivation

Since there are widely recorded challenges in phonological processing tasks in hearing children with language impairments, it can be assumed that similar challenges would also show up in the deaf population with ASL. However, only a few studies have been conducted on native signing deaf students suspected of having dyslexia or sign language impairment. In these studies, deaf students performed poorly on short-term sequential memory tests such as fingerspelling words, recalling sequences of items, and repeating ASL sentences [52]. Yet there is a paucity of knowledge regarding the role of ASL rhyme, rhythm, and phonological awareness in capturing dyslexia or sign language impairments early on. What complicates the inquiry is the prevalence of deaf children going through their early years without language access, which confounds the line between signed language impairment and language deprivation syndrome [53,54].

A language deprived person is defined as an individual who went through their first years without language access and has had structural changes in the brain as a consequence [55]. Multiple studies have looked into the ability to imitate and recite in deaf adults who experienced language deprivation. These deaf adults had difficulty signing along simultaneously to what was signed to them and struggled to recall ASL sentences verbatim [56,57,58]. Knowing that the challenges of phonological processing tasks remain with deaf individuals into adulthood, questions are raised about the type of specialized interventions that should be given to young deaf children already experiencing language deprivation. Whether interventions that incorporate ASL rhyme, rhythm, and phonological awareness can help counteract the effects of language deprivation are yet to be investigated.

The literature review of rhyme, rhythm, phonological awareness, engagement, and recitation prompt a wide range of research questions surrounding the role of ASL rhyme and rhythm in young deaf children’s development. To date, there is little qualitative or quantitative research on ASL rhyme and rhythm. Thus, any experimental study conducted to explore the relationship between ASL rhyme and rhythm and other developmental areas in children will be significant and bring new knowledge and discussion to the field. The purpose of this research was to examine the effects of ASL rhyme and rhythm on Deaf children’s engagement behavior and accuracy in recitation. The research questions were as follows:(1)What are the effects of rhyming and non-rhyming conditions of ASL stories on Deaf children’s engagement behavior?(2)What are the effects of rhyming and non-rhyming conditions of ASL stories on Deaf children’s accuracy in recitation?(3)What are the effects of handshape rhyme awareness instruction on Deaf children’s engagement behavior and accuracy in recitation?

## 2. Method

The effects of ASL rhyme and rhythm on deaf children’s engagement behavior and accuracy in recitation were examined through single subject design. Individual performance was analyzed using a visual analysis that looks at the level, trend, variability, immediacy of effect, and consistency of data patterns within condition and between phases. Group performance was investigated through the mean and mean difference between both conditions and phases. Variables such as vocabulary knowledge and language ability that may impact overall results were examined. Information derived from social validity questionnaires provided insight into the significance of this intervention.

### 2.1. Alternating Treatments Design

The alternating treatments design capitalizes on the benefits of single subject research by giving two or more treatments to the same individual and then documenting the effects on target behaviors [59,60]. The quick alternation of two different conditions allow for direct comparison between treatments, minimizing potential confounding factors. This design brings a greater understanding of how deaf children respond to two different stimuli—rhyming ASL stories and non-rhyming ASL stories. A functional relation between independent and dependent variables is established when there is consistent evidence of an effect at a minimum of three different points in time [61]. The students needed to demonstrate higher levels of engagement behavior during viewing and higher levels of accuracy in reciting the rhyming or non-rhyming condition of the ASL stories on at least three days’ worth of attempts. A higher standard for assessing a functional relation is at least 4 or 5 data points [62].

### 2.2. Participants and Setting

Upon obtaining approval from the Institutional Review Board at the University of Tennessee (UTK IRB-18-04313-XP), teacher, child, and family participants were recruited from an early childhood program at an ASL/English bilingual deaf school in the western region of the USA. Teachers interested in participating in the study were asked to sign consent forms. Interested families were asked to complete the packet containing a consent form, a family background questionnaire, and a social validity questionnaire. If a family did not give consent for their child to participate in the study, no data was collected from their child. No child was turned away from being able to participate in the study as long as they maintained regular attendance in school.

Ten deaf children between three and six years old with varying backgrounds in language level, race, gender, sex, disability, hearing status, familial hearing status, home language, and socio-economic status participated in this study. By the end of the study, only five students met What Works Clearinghouse (WWC)’s criteria of having four or more data points in each condition for stronger evidence of functional relations. The other five students missed some sessions due to illness, of pulled out for special services or off-campus appointments, which disqualified them from visual analyses. However, there was a special case with a deaf of deaf child, who did not meet the WWC criteria, but exhibited atypical language patterns. This child is included in the report for the purpose of extending our understanding of a possible case of sign language impairment. Table 1 lists characteristics for each student participant. The teacher participants were two preschool teachers and a prekindergarten teacher. Two teachers were native deaf signers and one teacher was hearing but fluent in ASL. The teachers’ teaching experience ranged from five to twelve years.

Preschool and prekindergarten classes had a daily routine following a schedule of activities. When it was ASL time, students were seated in a semi-circle facing a large Smartboard. To be consistent with the classroom routine, the teacher in each classroom, with the first author’s assistance, introduced either the rhyming or non-rhyming condition of ASL stories in a video format on the Smartboard to the whole class, including students not participating in the study. After collecting engagement data from the students via a camera latched to the Smartboard, the first author called them individually to a private space next to the classroom where they were recorded reciting the story. Prior to the study, the first author came to their classroom daily for a week and asked them to view and recite videos from their regular curriculum to get the students acquainted with the researcher, the camera, and the process.

## 3. Materials

The first author created a total of five ASL videos—four versions of two ASL stories and an ASL story for the preferred condition—for the intervention in this study. The first author used to be an ASL teacher for deaf preschoolers, taught ASL courses in universities, and is a co-founder of Hands Land, a company that develops ASL rhyme and rhythm for young children. These experiences contributed to the qualifications required in developing the materials for this study. Both versions were similar in vocabulary and basic semantic content, but some of the words were ordered differently to eliminate rhyme and rhythm in the non-rhyming version. Both versions had the same rate and inflection, were syntactically correct, and made sense semantically. The videos were reviewed for production similarity by the classroom teachers and a deaf colleague prior to implementation. They confirmed that the only difference in the videos was the ordering of words and the absence of rhythms in the non-rhyming story. (See Table 2 and Table 3 for ASL gloss of the rhyming and non-rhyming versions of an ASL story.) Each line in the rhyming version had the same handshape for all the signed words while each line in the non-rhyming version had different handshapes. For handshape rhyme awareness lessons, the first author developed a slideshow presentation with images of rhyming signed words and images of individual handshapes.

## 4. Measures

### 4.1. Baseline Assessments

The researcher-made picture vocabulary assessment consisted of printed images of the selected 22 out of 45 vocabulary words in ASL Story 1 and ASL Story 2 (See Appendix A, Figure A1). The vocabulary words were mouse, raccoon, rooster, zebra, deer, one, two, three, four, five, red, orange, yellow, green, blue, purple, worm, bison, whale, bird, shark, and skunk. While vocabulary knowledge was not the main focus of this study, knowing whether students already had previously developed a lexicon of the target words in the ASL stories could help explain their performance during the intervention. For the purpose of analysis, three subgroups were formed regarding students’ knowledge of the target words in the ASL stories: little (0–8 signed words), some (9–18 signed words), and most (19–23 signed words). The Visual Communication Sign Language (VCSL) scores were collected from classroom teachers. The VCSL checklist tracks young children’s sign language development and provides information on whether they are meeting age-appropriate milestones [63]. Two subgroups based on language abilities indicated in the VCSL checklist were formed for this study: typical and delayed. Students behind in language by two years or more were considered delayed and placed in the delayed subgroup.

### 4.2. Social Vailidity Questionnaire

The researcher-made social validity questionnaire was a 21-item Likert scale ranging from strongly disagree to strongly agree. There were six categories in the questionnaire: knowledge, experience and uses, implementation, language development, preference and skills, and recommendations. A few examples of the items were: “I was familiar with ASL rhyme and rhythm prior to this research”, “I have access to ASL rhyme and rhythm videos”, “Signing along with ASL rhyme and rhythm videos is easy for me”, and “ASL rhyme and rhythm videos are good resources for families”. The questionnaire in English or Spanish was sent to families through regular school-to-home communication.

### 4.3. Independent and Dependent Variables

ASL stories with rhyming and non-rhyming versions were the two treatment conditions used to measure students’ engagement behavior and accuracy in recitation. The four dependent variables were: nonverbal engagement (viewing), verbal engagement (imitating), words recited correctly, and words recited in the correct order. The viewing behavior in nonverbal engagement was defined as eyes on the screen or eyes on peers (if their peers were signing in imitation of the source material). The imitating behavior in verbal engagement was defined as signing along with the signer in the video or peers using signed words associated with the ASL story. Disengagement was defined as eyes off the screen, eyes off the signer, or signing words not associated with the ASL story. Disengagement by interruption was defined as teacher interruption, student interruption, or other external distractions interfering with the student’s ability to attend and/or engage with the independent variable. Words recited correctly were defined as repeating and signing aloud any words from the ASL story from memory, regardless of the sequence of words. Words recited in the correct order were defined as repeating and physically signing the words of the ASL story from memory in the correct sequence.

### 4.4. Data-Recording Procedures

A permanent product in the form of videotaping was used to collect data on engagement behavior and recitation data. The videos collected for engagement behavior were immediately reviewed after each intervention session and a 5-second partial interval data recording procedure was used to indicate if the student was engaged or disengaged. The final metric was calculated by dividing the total number of 5-second interval engagement behaviors by the total intervals measured during a viewing session (*n* = 26–32).

Students’ recitation, which took place individually in a private room, was video recorded and measured using event recording procedures. The first part of the analysis awarded a point for each word recited correctly from the ASL story, regardless of the sequence of the signed words. The second part of the analysis gave a point for each word recited in the correct order. The number of words signed correctly and the number of words signed in the correct order in the rhyming condition and the non-rhyming condition were analyzed and compared to determine the preferred condition in increasing accuracy in recitation.

### 4.5. Procedural Integrity

Teachers involved in this study received one hour of consultation on administering the whole class intervention with integrity. They were asked to stick with their routine of calling students to sit down in a U-circle, asking them to be ready to view a story on screen, clicking ‘play,’ and then staying behind students. The first author was present at all sessions and provided immediate feedback when teachers did not achieve fidelity. In 42 sessions, 155 teacher behaviors out of 168 of the total amount of planned teacher behaviors were successfully executed for a total of 92% procedural integrity. 

The second part of the intervention did not include classroom teachers, as students were individually called by the first author to the conference room to re-watch the ASL story on a laptop and recite it to camera. All of the 194 recitation sessions met the procedural integrity with 100% fidelity.

### 4.6. Inter-Rater Reliability

The reliability of the student data was established through the inter-rater agreement of 90% accuracy or above by the first author and another deaf colleague with a doctoral degree who is also fluent in ASL. If there were disagreements, both observers viewed the video again and discussed their observations until an agreement was reached. Thirty percent of the task engagement data and 35% of the recitation data were double scored, and interrater reliability was 95% and 96% respectively.

### 4.7. Visual Analysis

Insights from meta-analyses along with WWC’s criteria for procedures and standards on alternating treatments design informed the decision to use visual analysis and the total mean and mean difference in this study [62,64,65,66,67]. Visual analysis of single subject data addresses whether behavior changed and if that change came from the intervention. Six indicators were used to evaluate within-phase and between-phase data patterns to judge the extent of the effects of the intervention: (a) level, (b) trend, (c) variability, (d) immediacy of the effect, (e) overlap, and (f) consistency of data patterns across similar phases [68]. Examination of within- and between data patterns using these six indicators informed the existence of causal relation and the strength of its evidence [69].

### 4.8. Total Mean and Mean Difference

The total mean and mean difference procedures were conducted for the whole group to compare the effects of rhyming and non-rhyming conditions. Non-parametric statistical tests, the Wilcoxon Signed-Rank Tests, were performed to compare baseline means across rhyming and non-rhyming conditions. There was no basis for prediction at the baseline, making it exploratory. The treatment means across rhyming and non-rhyming conditions were predicted to be significant. Finally, a comparison of the preference phase and treatment means of the same type was predicted to be non-significant.

## 5. Procedure

Prior to initiating the baseline sessions, family questionnaires were sent to families by putting the documents in the students’ backpacks and then collecting them when they were sent back to school—which was the routine home-to-school communication. The first author administered the researcher-developed picture vocabulary assessment to each student individually in a private room at the educational setting. The first author pointed to the image and then signed: “WHAT–THIS?” If the student did not provide a signed word, a pause of five seconds was given before proceeding to the next picture. If the student signed, “DO NOT KNOW,” the first author proceeded to the next image. If the student responded with an incorrect signed word, the first author would again point to the image and sign “WHAT–THIS?”. This provided the student another opportunity to look at the picture and correct their mistake. If they did not provide the correct signed word on their second attempt, then the first author proceeded to the next image. Information on students’ present language levels (VCSL) was collected from the classroom teachers.

There were four phases of interventions in this study: (1) baseline, (2) handshape rhyme awareness intervention, (3) alternating treatments, and (4) preference. After two weeks of baseline alternating treatments sessions in both conditions, there was no observable bifurcation [70] in students’ engagement behavior and accuracy in recitation. Thus, baseline data confirmed the concern of students lacking handshape rhyme awareness, and this evidence led to the decision to implement the handshape rhyme awareness intervention. The added condition was two 20-minute lessons on handshape rhyme awareness given by the first author. The lessons paralleled what was found in lessons that teach rhyme recognition in spoken language. Activities in the lesson involved asking students: “What same handshape was used for all of the signed words in the video?” The first author encouraged students to pay attention and notice the handshape patterns in signed words. Following the handshape rhyme awareness intervention, alternating treatments of the second story took place to determine the effects of the intervention on engagement behavior and accuracy in recitation. After the preferred treatment was identified through visual analysis, the least effective condition was discontinued, and the more effective treatment was replicated on subsequent days using a new story (See Figure 1 for the intervention schedule).

## 6. Results

### 6.1. Engagement

The mean of total percentage of engagement occurrence intervals in the baseline was 38% imitating and 54% viewing in the rhyming condition, and 26% imitating and 61% viewing in the non-rhyming condition. The mean of total percentage of engagement occurrence intervals in the alternating treatments phase was 44% imitating and 32% viewing in the rhyming condition, and 22% imitating and 58% viewing in the non-rhyming condition. The mean of total percentage of engagement occurrence intervals in the preference phase was 45% imitating and 34% viewing in the rhyming condition. There were no statistically significant differences on the Wilcoxon Signed-Rank Test for baseline (Z = −0.81, *p* < 0.42), treatment (Z = −0.67, *p* < 0.5), or preference phase compared to treatment (Z = −0.94, *p* < 0.35). The same was true for imitation for baseline (Z = −1.75, *p* < 0.08), treatment (Z = −1.83, *p* < 0.07), and preference phase compared to treatment (Z = −0.41, *p* < 0.69); however, total means seem to show imitation to be happening in the rhyming condition more than the non-rhyming condition, especially during the treatment phase (M = 43.8 in rhyming condition compared to M = 22.4) (See Figure 2).

Students’ picture vocabulary assessment scores and their language skills were examined as variables that may influence engagement behavior. There were two subgroups of students’ knowledge of the target words in the ASL stories: some (9–18 signed words) and most (19–23 signed words). The two subgroups of delayed and typical language skills were determined based on the VCSL checklist. The two students that knew the most vocabulary were the same two students that had age-appropriate language skills. Similarly, the three students that knew some vocabulary had language delays. The most vocabulary knowledge and typical language subgroup demonstrated higher total means in imitation in both conditions (74% in the rhyming condition, 57% in the non-rhyming condition) than the some vocabulary knowledge and language delayed subgroup (21% in the rhyming condition, 3% in the non-rhyming condition) (See Figure 3).

### 6.2. Recitation

The mean percentages of words signed correctly in rhyming and non-rhyming conditions in each phase and across phases are presented in Figure 4. During the baseline, the mean percentage of words signed correctly in the rhyming condition was 52% and in the non-rhyming condition it was 41%. After handshape rhyme awareness intervention was given, the mean percentage of words signed correctly in the rhyming condition was 69% and in the non-rhyming condition it was 36%. In the preference phase, the mean percentage of words signed correctly in the rhyming condition was 67%. Wilcoxon Signed-Rank Tests indicated that the baseline and treatment means for words signed correctly in the rhyming conditions were statistically significantly higher than the non-rhyming conditions, Z = −2.03, *p* < 0.04; Z = −2.02, *p* < 0.04, although the total mean in the treatment (M = 68.8 in rhyming condition compared to M = 36.0) shows more substantial difference than at baseline (M = 52 in rhyming condition and M = 40.8 in non-rhyming condition). Furthermore, as predicted, the preference condition ranks were not statistically significantly different compared to the treatment condition ranks, Z = −0.41, *p* < 0.67.

The mean percentages of words signed in the correct order in rhyming and non-rhyming conditions in each phase and across phases are demonstrated in Figure 5. In the baseline, the mean percentage of words signed in the correct order in the rhyming condition was 38% and in the non-rhyming condition it was 32%. After the handshape rhyme awareness intervention, the mean percentage of words signed in the correct order in the rhyming condition was 64% and in the non-rhyming condition it was 28% during the alternating treatments phase. In the preference phase, the mean percentage of words signed in the correct order in the rhyming condition was 62%. Wilcoxon Signed-Rank Tests performed for words signed in correct order show the means for rhyming condition in the treatment phase are statistically significantly higher than those in the non-rhyming condition, Z = −2.02, *p* < 0.04, while no significant differences were detected at baseline, Z = −4.05, *p* < 0.69, or between the preference and treatment conditions, Z = −0.37, *p* < 0.72.

In the most vocabulary and typical language subgroup, the mean percentage of words signed in the correct order was 80% in the rhyming condition and 39% in the non-rhyming condition. In the some vocabulary and delayed language subgroup, the mean percentage of words signed in the correct order was 25% in the rhyming condition and 18% in the non-rhyming condition. Total means seem to show more words signed in the correct order in the rhyming condition than the non-rhyming condition in both subgroups (See Figure 6).

### 6.3. Visual Graphs

Visual graphs showing student performance in the rhyming and non-rhyming conditions with data overlapped for visual comparison are presented for words signed correctly and words signed in the correct order during recitation. The order of students is listed based on age, starting with the youngest, with an exception for Lacey who did not meet the WWC standard of having four or more data points for each condition in each phase. Lacey’s visual graph is included at the end for the purpose of exploring a possible case of sign language impairment (See Figure 7, Figure 8, Figure 9, Figure 10, Figure 11, Figure 12, Figure 13, Figure 14, Figure 15, Figure 16 and Figure 17). Most students performed relatively similarly when it came to accuracy in recitation in both conditions in the baseline, with the rhyming condition being slightly superior. After receiving handshape rhyme awareness intervention, most students’ accuracy in recitation increased in the rhyming condition during alternating treatments and preference phases.

#### 6.3.1. Daya

Daya was four years old and had language delays and some vocabulary knowledge. Daya came from non-signing hearing parents and had been learning ASL for a little over a year. After receiving handshape rhyme awareness intervention, and upon introduction of the alternating treatments phase, Daya’s level, trend, and variability between both conditions in the alternating treatments were similar with higher mean percentages in the rhyming condition than the non-rhyming condition (words signed correctly: 14% higher; words signed in the correct order: 14% higher). There was a consistent and small separation of the data paths between both conditions, with the rhyming condition remaining higher over the non-rhyming condition.

#### 6.3.2. Yair

Yair was four years old and came from a deaf family. Yair was delayed in language by two years and had some vocabulary knowledge. Yair’s level, trend, and variability between both conditions in the alternating treatments phase were similar with the mean percentage of the rhyming condition being 12% higher than the non-rhyming condition for words signed correctly and 15% higher than the non-rhyming condition for words signed in the correct order. There was a separation in data paths between both conditions of a small to moderate magnitude with the rhyming condition being superior.

#### 6.3.3. Giada and Jaslene

Giada and Jaslene were five-year-old participants who had early access to ASL, age-appropriate language skills, and high vocabulary knowledge. During the baseline phase, both Giada and Jaslene appeared to be oblivious to the existence of handshape rhymes and used the same approaches in their effort to memorize and recite the entire story in rhyming and non-rhyming conditions. For example, they would pause when they could not remember what words came next and demonstrate a “thinking face” as they waited for the words to come to mind. They did not rely on handshape rhymes to clue them as to what come next in the story. They both started the first session of the baseline phase with 56% and 8% words signed in the correct order in the rhyming condition, respectively, and 38% and 0% in the non-rhyming condition, respectively. The accuracy of their recitations in the last session was close to 100% in both conditions. During the handshape rhymes intervention, Giada and Jaslene were shown the rhyming condition of ASL Story 1 and had the handshape rhymes pointed out—both Giada and Jaslene’s eyes widened, and their mouths stood agape. Giada put hands to face as if to say: “Why didn’t I see it before?!” During the alternating treatments phase, Giada and Jaslene’s behavior changed: they immediately pointed out the handshape rhymes in the rhyming condition both during and after viewing and made accurate comments about them. Then, Giada and Jaslene made a big jump in their first session of alternating treatments and recited the rhyming condition of the ASL Story 2 with 92% and 80% of words in the correct order, respectively. Giada and Jaslene’s level, trend, and variability in data paths between both conditions in words signed correctly and words signed in the correct order demonstrated a consistent and large magnitude of separation with the rhyming condition having higher mean percentages over the non-rhyming condition (words signed correctly: 73% and 48% higher, respectively; words signed in the correct order: 77% and 56% higher, respectively).

#### 6.3.4. Lexie

Lexie was six years old—the oldest participant in this study—and was adopted from another country two years ago. Lexie came to the United States without any language or lexical vocabulary. After two years of intensive language immersion and support provided by their adoptive deaf parents and teachers of the deaf, Lexie’s language skills in ASL grew to the three-year-old level according to the VCSL. Lexie’s trend and moderate variability between both conditions in the alternating treatments were comparable with the mean percentage of the rhyming condition, being 17% higher than the non-rhyming condition for words signed correctly and 9% higher than the non-rhyming condition for words signed in the correct order. There was a separation in data paths between both conditions of a small to moderate magnitude with the rhyming condition being higher for words signed correctly.

#### 6.3.5. *Lacey 

*Lacey had three data points in the rhyming condition during the alternating treatments phases, which did not meet the WWC criteria for inclusion in this study. However, their recitation data is still included here for the purpose of expanding our understanding of a possible case of sign language impairment. Lacey was almost four years old at the time of the study. Lacey came from deaf parents, had age-appropriate language skills, and high vocabulary knowledge. Lacey did not sign any words in the correct order in four out of five rhyming sessions during the baseline. This trend continued in the alternating treatments phase with 0% of words signed in the correct order for three consecutive sessions. Lacey produced more words in the correct order in the non-rhyming condition with a range of 17–21% of words signed in the correct order in the baseline, and a range of 0–15% words signed in the correct order in the alternating treatments phase (See Figure 17).

### 6.4. Social Validity Questionnaire

Although only five students met the criteria of having four or more data points in each condition and phase, there were a total of ten students participating in this study. Eight out of ten caregivers, five hearing and three deaf, returned the social validity questionnaire. When asked if they knew how to make rhymes in ASL, four people “agreed,” two people “disagreed,” one person “strongly disagreed,” and one person was “uncertain.” When asked if they were familiar with ASL rhyme and rhythm prior to this research, five people “agreed,” three people “disagreed” and one person “strongly disagreed.” When asked if they had access to ASL rhyme and rhythm videos at home, one person “strongly agreed,” four people “agreed,” one person “disagreed,” and two people were “uncertain.” When asked if signing along with ASL rhyme and rhythm videos was easy for them, six people “agreed,” and two people were “uncertain.” When asked if signing ASL rhyme and rhythm without videos was easy for them, three people “agreed,” and five people “disagreed.” When asked if they thought ASL rhyme and rhythm were a good way for families to learn sign language, two people “strongly agreed,” four people “agreed,” and two people were “uncertain.” When asked if they thought ASL rhyme and rhythm videos were good resources for families, two people “strongly agreed,” and six people “agreed.” A caregiver left a comment on the questionnaire, “Anytime spent communicating with your child is particularly important for bonding. Using fun ASL rhymes and rhythms would only enhance this experience.” Another caregiver also wrote a note on the questionnaire, “Rhyming in groups I think might be analogous to singing in chorus—a social activity.” Caregivers’ overall responses were either positive or uncertain as some were familiar with ASL rhyme and rhythm while others did not know ASL nor had any exposure to this practice.

## 7. Discussion

Research on young hearing children shows that engagement, imitation, rhyme awareness, and recitation play an integral role in language development. This study explored whether ASL rhyme and rhythm had similar results in deaf children by comparing the effects of rhyming and non-rhyming conditions of ASL stories after giving explicit instruction in handshape rhyme awareness. The results of the group means did not indicate a significant impact on engagement. However, significant gains could be found in group means of accuracy in recitation.

### 7.1. Engagement and Imitation

There was a difference in the number of intervals deaf children spent imitating compared to viewing ASL stories with and without rhyme and rhythm, but this was not significant. What is remarkable about this outcome is that deaf children engaged in imitating behavior in both conditions on their own without any instruction or modeling from adults. This spontaneous and naturalistic behavior while attending to stories, including those with rhyme and rhythm, shows how deaf children are not any different from hearing children in their imitating behaviors [71]. Furthermore, deaf children with language delays in this study imitated substantially less than deaf children with typical language skills. This is in align with research that found hearing children with language delays demonstrated fewer imitating behaviors while listening to songs [25]. Although deaf children with language delays imitated less than deaf children with age-appropriate language skills in this study, it is important to note that they still imitated more while viewing ASL rhyme and rhythm than with ASL stories without rhyme and rhythm. It may be that deaf children’s brains are naturally seeking phonological patterns, which affirms the hypothesis of Petitto et al. [15] on the significance of sign phonology in language development. The alluring and fun nature of rhyme and rhythm seem to elicit more imitating behaviors in hearing and deaf children alike.

### 7.2. Recitation and Phonological Awareness

This study departed from past studies by shedding light on whether deaf children are able to independently recognize rhymes and recite them. Although their performances were slightly better in the rhyming condition at the baseline, deaf children did not appear to recognize the existence of rhymes when exposed to ASL rhyme and rhythm and struggled with reciting words in the correct sequential order. Their lack of handshape awareness was incongruent with the literature on rhyme awareness development in the population of young hearing children [18]. This finding was somewhat unsurprising considering that deaf children have extremely limited exposure to and experience with ASL rhyme and rhythm. The first author’s experience providing professional development on ASL rhyme and rhythm reinforces this phenomenon. During the training sessions, many of the professionals at the schools serving the deaf children lacked rhyme awareness themselves. Additionally, classroom teachers in this study confirmed that their students had extremely limited exposure to ASL rhyme, rhythm, and phonological awareness. The lack of handshape rhyme awareness may help explain why they did not initially respond to rhymes in the ASL story.

Deaf children may need to first master the prerequisite skills of handshape identification, handshape categorization, and rhyme knowledge in order to have heightened awareness and appreciation of the features found in ASL rhyme and rhythm. Without these foundational skills, children in this study did not think of using handshape rhymes as a tool to support the sequential memory that is required for the task of recitation. Multiple studies on hearing children have stressed supplementing song recitations with phonological awareness activities for better outcomes in engagement and recitation [20,22,23,72]. Overarching theoretical postulations exist on the importance of having the skills to recognize rhymes for enhanced ability in remembering vocabulary [73], word pairs [74], sequences [34] and stories [33]. What can be learned from this study is that when deaf children have minimal, if any, exposure to ASL rhyme and rhythm, they are not given opportunities to develop rhyme awareness. Without rhyme awareness, they are oblivious to the existence of rhymes in ASL songs or stories. Two 20-minute interventions to teach deaf children to recognize handshape rhymes produced mixed results in this study. Deaf children with age-appropriate language picked up on the new skill quickly, while there seem to be variations in language delayed deaf children’s performance. Their overall performance was not unlike the literature on variations in hearing children with disabilities’ acquisition of rhyme awareness with interventions that span months, or even years [27,28].

### 7.3. Student Performance

Giada and Jaslene’s commensurable age, language abilities, vocabulary knowledge, and the similarities in their performance during the baseline, alternating treatments, and preference phases make a useful case study that parallels the literature on typically developing hearing children. Their ability to successfully recite ASL stories (whether they had rhymes or not) is an age-appropriate skill [75]. While viewing, they demonstrated more imitating behavior than they did in the baseline. This marked gain in imitation is important, as the function of imitation in young children may be related to processing linguistic input from the environment, which helps with memory and furthers their understanding of language. It is argued that children imitate only the phonological information that they can perceive and understand well enough to repeat [76]. While reciting, Giada and Jaslene clearly relied on handshape rhymes as clues to guide them in remembering what signed words should come next. When Giada and Jaslene gave the wrong signed word, they quickly caught their mistake because the signed word did not rhyme with the previous signed word. This behavior shows Giada and Jaslene specifically thinking about the linguistic feature of handshape rhyme and reflecting upon their own language production, judging the handshape in their signed word as an error to be corrected. This type of self-correction demonstrates metalinguistic awareness. The literature states that hearing children as young as three years old can possess metalinguistic skills that include self-correcting behaviors in language output, and that children with language impairments often struggle in this area [77]. Since interrelationships exist between metalinguistic awareness, phonological awareness, and language abilities [78], Giada and Jaslene’s successes in recitation in the rhyming condition could be attributed to these factors.

Daya, on the other hand, rarely imitated any signed words while viewing ASL stories in both conditions across phases. Accuracy in recitation was also low. This performance could be explained by overlapping factors related to language deprivation impacting phonological processing and language development. Multiple studies have looked into deaf adults who experienced language deprivation during their early years and their ability to imitate and recall. These deaf adults had difficulty signing along simultaneously to what was signed to them and struggled to recall sentences verbatim [56,57,58]. It appears that the language processing gap possibly stemming from language deprivation already begins to widen in four-year-old students of non-signing parents like Daya. Similar issues could be seen in Lexie who also experienced language deprivation during the early years.

As the oldest participant in this study, Lexie’s case of language deprivation was the most extreme, not having had any language access until four years old. The fact that Lexie had complete access to language at home in addition to being placed in an ASL-rich environment for two years made a difference in their ability to engage with language and recite. Lexie having progressed from a child with no language to a child who could perform as well as some four-year-old deaf children within a two-year period is encouraging. This informs us that having complete access to language through ASL at school and at home can build skills to assist with language development such as imitation and recitation.

The subpar performance of Lacey, a native signing deaf student raised some questions regarding the understudied phenomenon of dyslexia, signed language impairment, or ADHD in the population of deaf children from signing families. By documenting deaf children’s imitating behaviors and their ability to recite ASL rhyme and rhythm in this study, it was possible to recognize more clearly the language processing gaps in certain students. Because Lacey comes from a deaf family, has had access to ASL since birth, and had high vocabulary knowledge and age-appropriate language skills, the results seemed to be an anomaly for someone of this particular background. When asked to recite the ASL rhyme and rhythm, Lacey was not able to sign most of the words accurately nor in the correct order. In a follow-up interview with teachers, they said they noticed something was amiss with Lacey’s language and academic performance. In class lessons where children were expected to recall an ASL story, Lacey had a tendency of not following the correct sequence. Lacey also struggled with remembering the sequence of numbers. The teachers attributed the weakness to the child’s “free spirit” personality and relatively slow social-emotional development.

In spite of the fact of having been signing since birth, Lacey struggled the most of all students with recitation, demonstrating challenges with language processing tasks pertaining to working and sequential memory. However, classroom teachers did not raise this as a red flag for sign language impairment or ADHD. They were not concerned because Lacey was able to produce ASL sentences independently and engage in meaningful turn-taking conversations, unlike peers who were much more delayed due to language deprivation. In other words, issues with language processing in this child were overshadowed by classmates’ even weaker skills. While this is an understandable reality, not raising a red flag for dyslexia or language impairment meant that specialized attention is not being given to Lacey’s unique language needs.

Interventions similar to the ones used in this study could provide opportunities for professionals to attend to whether language processing skills are lacking, especially in deaf children who are native users of ASL. These kinds of intervention are also used to alert to the possibilities of dyslexia, language impairment, and ADHD in hearing children. There are well documented difficulties in phonological processing tasks such as recognizing words that sound the same and reciting rhyme and rhythm [79]. Whether this is paralleled to sign language impairments in deaf children from signing parents requires further investigation.

### 7.4. Social Validity

The social validity of exposing deaf children to rhyming ASL stories was explored through a questionnaire. While deaf parents were enthusiastic and asked for more resources, most hearing parents had limited knowledge of and were uncertain about this practice. A hearing parent’s note on the questionnaire illustrates their uncertainty, “Rhyming in groups I think might be analogous to singing in chorus—a social activity.” A deaf parent wrote that ASL rhyme and rhythm are fun to create at home. Deaf children are highly motivated to imitate, and the experience often ends with everyone bursting into laughter. This comment paints a picture of the high social and cultural importance of this practice for deaf families. All parents, hearing and deaf, agreed that ASL rhyme and rhythm videos are good resources for families. Most parents said it would be hard to expose their deaf children to ASL rhyme and rhythm at home if there were no videos available.

Teachers, like the deaf parents, spoke highly of the role of ASL rhyme and rhythm in fostering language development and lamented over the lack of resources. They did not feel knowledgeable and confident enough to sign ASL rhyme and rhythm on their own in their instruction. A teacher wrote in the questionnaire that they could incorporate ASL rhyme and rhythm, but limited resources create a stumbling block. Six months after the study took place, the principal requested the first author to return and give an all-day professional development to the department on implementing ASL rhyme and rhythm in their classrooms. During the professional development, teachers commented on witnessing the ways ASL rhyme and rhythm have promoted repetitions and patterns, memorization, creativity and play, metalinguistic awareness, prediction, humor, family-child bond, and turn-taking skills in their classes. Issues of resources and training being scarce present major barriers to exposing deaf children to ASL rhyme and rhythm at home and in school.

### 7.5. Limitations and Future Directions

There are limitations with the ASL assessments currently available to the public [54]. In this study, the Visual Communication Sign Language (VCSL) checklist was selected to approximate students’ language abilities according to developmental milestones. While this assessment was helpful in determining language skills (i.e., typical and delayed) which was important to understanding how language impacted students’ engagement behavior and accuracy in recitation, the assessment does have limitations. The VCSL checklist was normed on a small sample of children, which calls into question the ability to accurately quantify language delay by years. Notwithstanding, it is one of the few tools that track young children’s language milestones in ASL. The information it provided for this study was valuable. Future studies would benefit from the inclusion of assessment tools that can more precisely capture deaf children’s ASL phonological awareness and language skills.

Information extrapolated from this study indicates that not all deaf children have abundant exposure and experience with language—let alone ASL rhyme, rhythm, and phonological awareness—and this impacts their language processing abilities. There are still many unanswered questions. What types of specialized interventions in ASL are effective in closing language gaps? Is training in phonological processing tasks such as imitation and recitation suitable for deaf children as young as three and four years old? What is the role of ASL rhyme and rhythm in these interventions? More specifically, do they need to learn how to successfully imitate, recognize rhymes, and recite ASL rhyme and rhythm as part of the building blocks towards stronger language foundation and emergent literacy skills? Then, there is the question of the number of interventions needed to successfully build these skills. A comprehensive evaluation of ASL phonological awareness activities over a period of time across deaf learners is needed to thoroughly investigate the effects on language and emergent literacy development. Whether ASL rhyme, rhythm, and phonological awareness can also be used to identify potential cases of sign language impairments need to be examined.

## 8. Conclusions

The review of the literature provides clarity into the large gaps in our empirical knowledge of the role of ASL rhyme, rhythm, and phonological awareness in facilitating language and emergent literacy in young deaf children. A body of research has been built to affirm the significance of exposing hearing children to rhyme and rhythm supplemented with training in phonological awareness for successful literacy development. Yet interventions that incorporate deaf cultural and linguistic approaches using ASL are novel to most classrooms that serve deaf children. The results of this study have implications for potential positive change at the individual, cultural, educational, and societal levels. At the individual level, the results of this study inform the field that certain interventions such as imitation training, handshape rhyme awareness and recitation of rhyming ASL stories may have a favorable impact on deaf children’s language processing abilities, which are directly linked to critical emergent literacy skills in the population of young hearing children [80]. At the cultural level, deaf community members have long offered culturally-rich linguistic models through ASL storytelling, poetry, rhyme and rhythm, and language games. When the deaf community sees their linguistic and cultural capital [81] become an important part of deaf students’ experience in schools, this may lead to a greater understanding, appreciation, and validation of ASL literature—including the genre of ASL rhyme and rhythm. Should this occur, there may be a shift in deaf children’s relationship with language and music, making their experiences more deaf-centric and empowering. At the educational level, this study casts light on how teachers are often untrained in ASL rhyme, rhythm and phonological awareness. Without proper systemic support, deaf children are being deprived from accessing essential language exposure and experience that hearing children have. The lack of proper interventions may have an impact on deaf children’s language and literacy, stalling their ability to maximize their linguistic potential. Educators can use these data to advocate for more training in culturally and linguistically responsive approaches. The findings from this study also provide a foundation for future research to explore interventions that are not only “new and better,” but also specifically geared to bilingual learners such as deaf children who are primed for the benefits of metalinguistic awareness and linguistic transfers. Considering that this study has sought to address the gaps in pedagogy due to long-standing systemic barriers towards the acceptance of deaf cultural and linguistic practices, outcomes might also have implications at the societal level. This new knowledge about the role of ASL rhyme and rhythm in early childhood development may propel society to take steps towards generating a paradigm shift in uplifting deaf pedagogy.

## Figures and Tables

**Figure 1 children-07-00256-f001:**
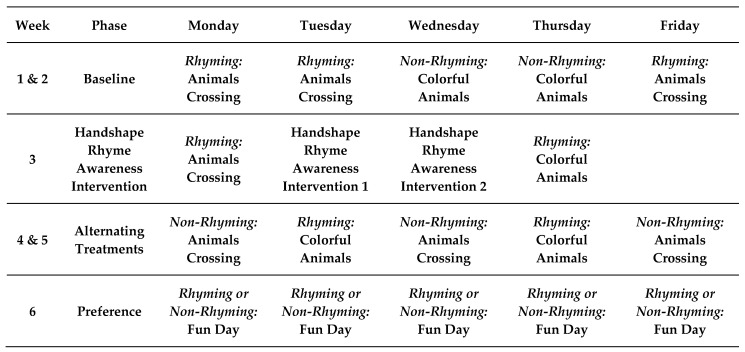
Intervention Schedule.

**Figure 2 children-07-00256-f002:**
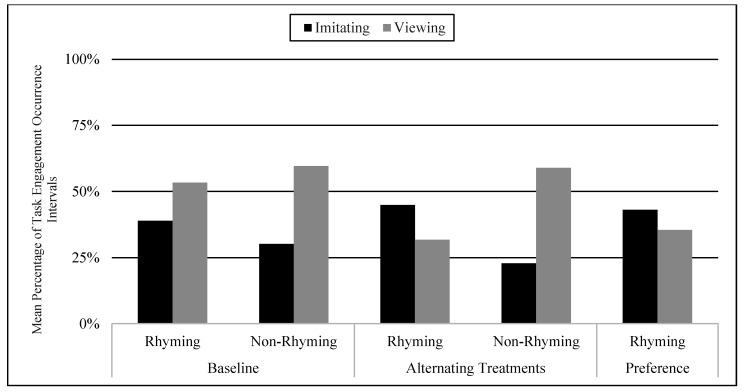
Engagement: Mean Percentage of Task Engagement Occurrence Intervals in Viewing and Imitating Behaviors.

**Figure 3 children-07-00256-f003:**
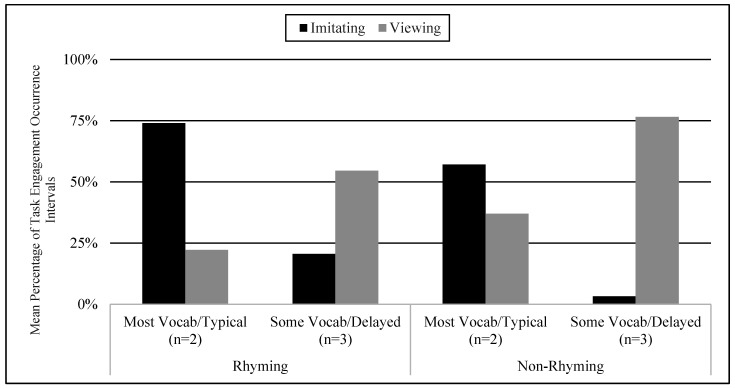
Mean Percentage of Task Engagement Occurrence Intervals in Viewing and Imitating Behaviors Across Phases in Some Vocabulary/Delayed and Most Vocabulary/Typical Groups.

**Figure 4 children-07-00256-f004:**
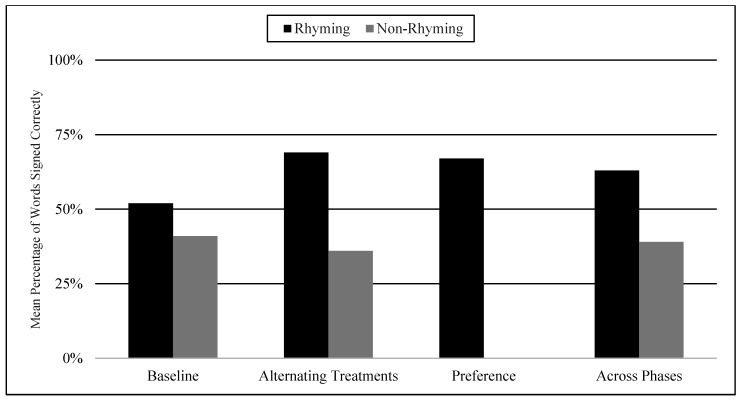
Mean Percentage of Words Signed Correctly in Rhyming and Non-Rhyming Conditions in Each Phase and Across Phases.

**Figure 5 children-07-00256-f005:**
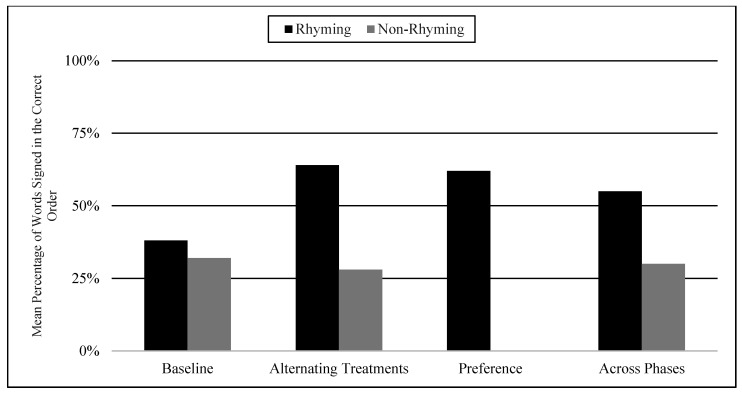
Mean Percentage of Words Signed in the Correct Order in Rhyming and Non-Rhyming Conditions in Each Phase and Across Phases.

**Figure 6 children-07-00256-f006:**
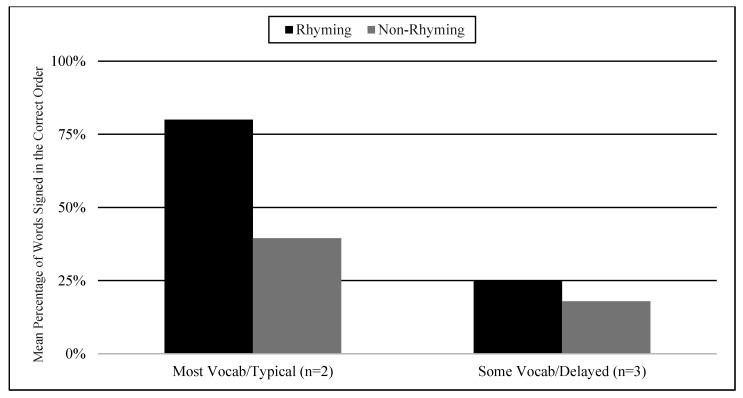
Mean Percentage of Words Signed in the Correct Order Across Phases in Some Vocabulary/Delayed and Most Vocabulary/Typical Language Groups.

**Figure 7 children-07-00256-f007:**
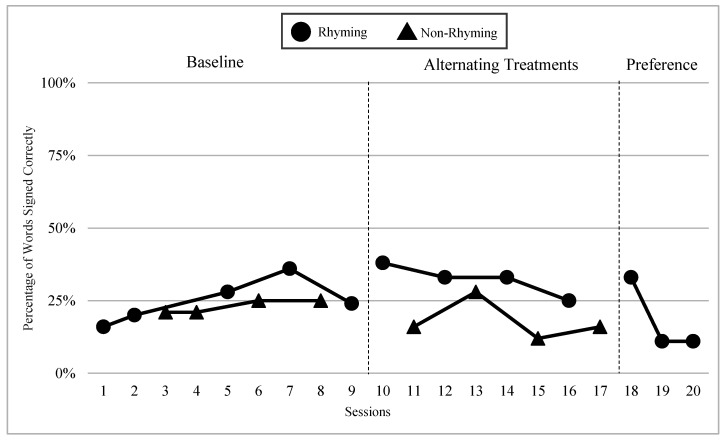
Daya’s Percentage of Words Signed Correctly in Recitation.

**Figure 8 children-07-00256-f008:**
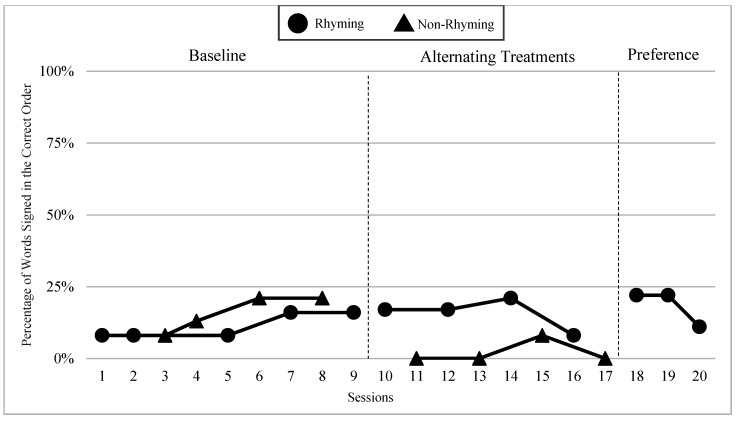
Daya’s Percentage of Words Signed in the Correct Order in Recitation.

**Figure 9 children-07-00256-f009:**
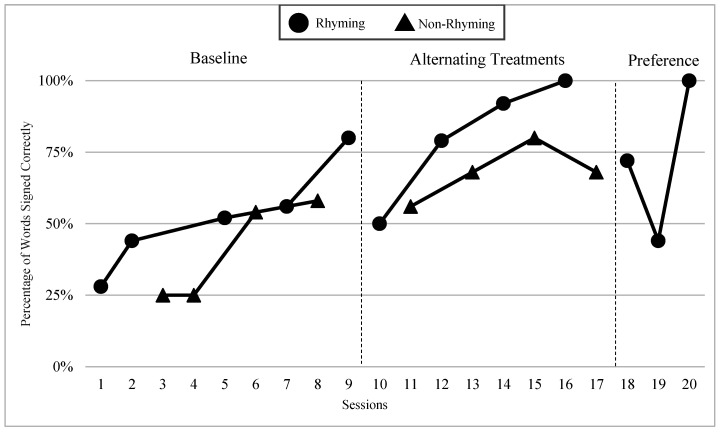
Yair’s Percentage of Words Signed Correctly in Recitation.

**Figure 10 children-07-00256-f010:**
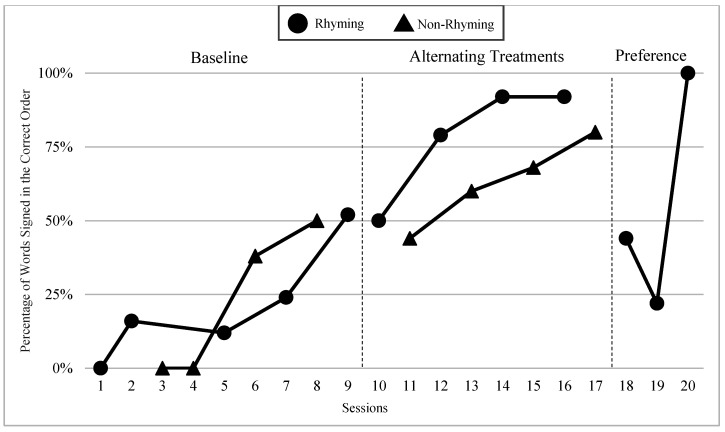
Yair’s Percentage of Words Signed in the Correct Order in Recitation.

**Figure 11 children-07-00256-f011:**
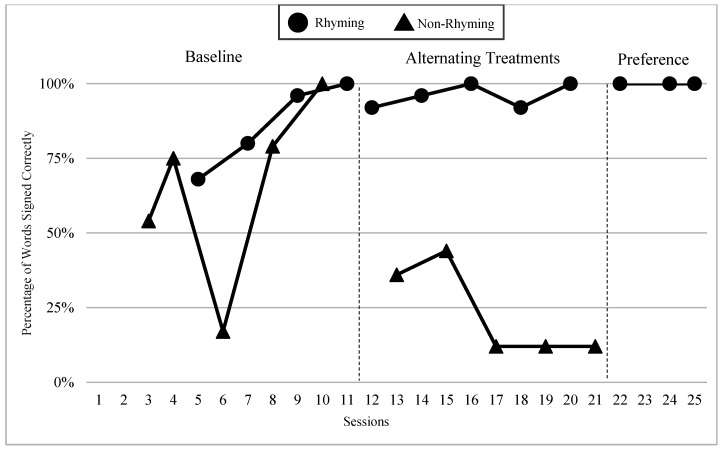
Giada’s Percentage of Words Signed Correctly in Recitation.

**Figure 12 children-07-00256-f012:**
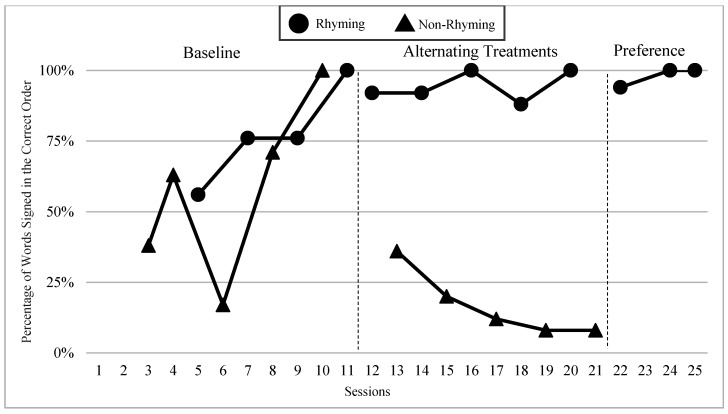
Giada’s Percentage of Words Signed in the Correct Order in Recitation.

**Figure 13 children-07-00256-f013:**
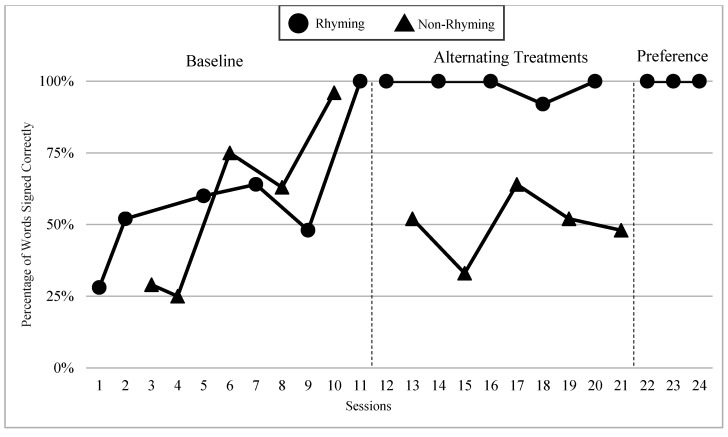
Jaslene’s Percentage of Words Signed Correctly in Recitation.

**Figure 14 children-07-00256-f014:**
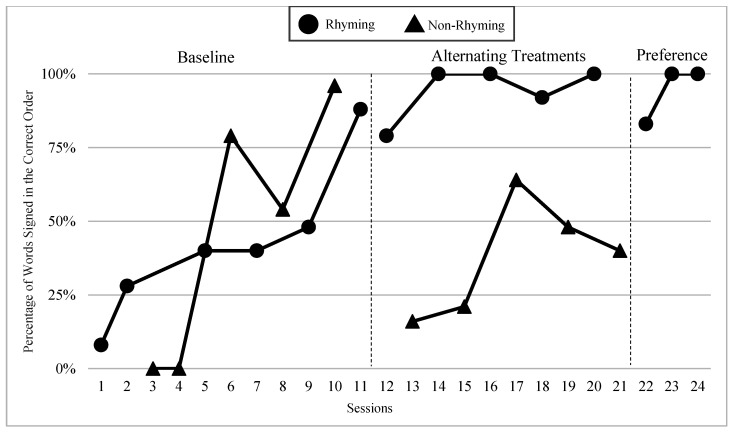
Jaslene’s Percentage of Words Signed in the Correct Order in Recitation.

**Figure 15 children-07-00256-f015:**
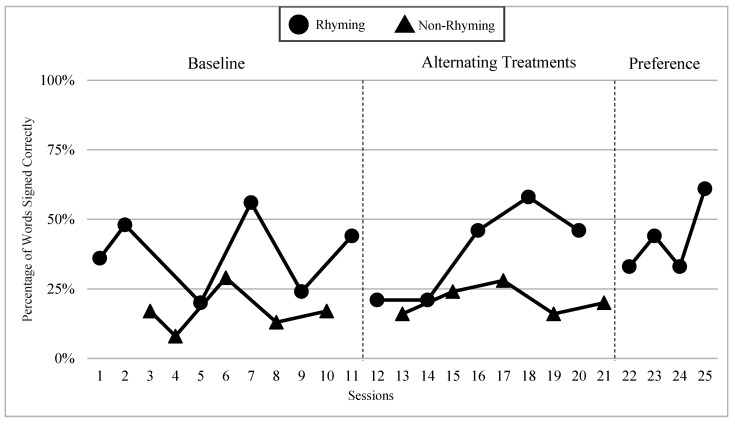
Lexie’s Percentage of Words Signed Correctly in Recitation.

**Figure 16 children-07-00256-f016:**
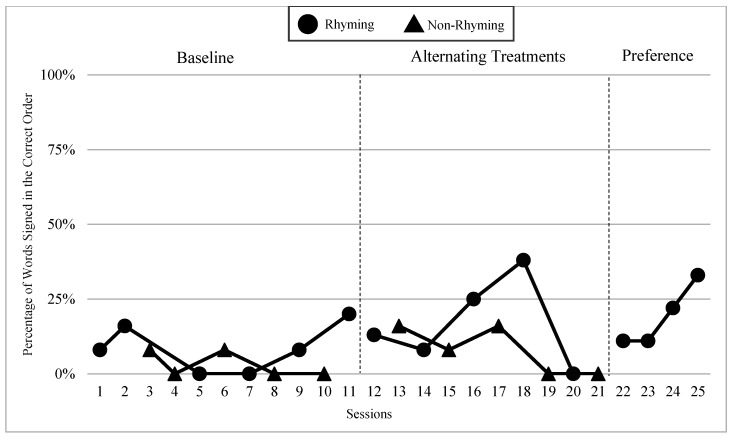
Lexie’s Percentage of Words Signed in the Correct Order in Recitation.

**Figure 17 children-07-00256-f017:**
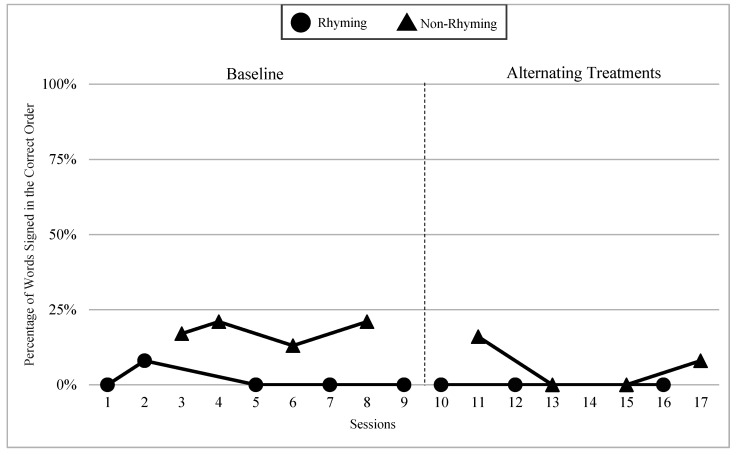
*Lacey’s Percentage of Words Signed in the Correct Order in Recitation.

**Table 1 children-07-00256-t001:** Student Participants’ Characteristics.

Name	Class	Age	ASL	Vocab.	VCSL	Sex	Race	Disab.	P.H.S.	H.L.
Daya	P.S.	4.6	3 years old	10/23	2.7	F	White	None	H + H	Eng.
Yair	P.S.	4.10	Birth	16/23	2.4	M	Asian	None	D + D	ASL
Giada	P.K.	5.7	4 months old	21/23	4.5	M	White	None	H + H	ASL & Eng.
Jaslene	P.K.	5.10	Birth	21/23	4.5	M	Mixed	None	D + D	ASL
Lexie	P.K.	6.5	4 years old (adop.)	14/23	2.8	F	Asian	None	D + D	ASL & Sign. Lang.
* Lacey	P.S.	3.10	Birth	19/23	3.7	F	Asian	None	D + D	ASL

Notes: Names are pseudonyms to maintain confidentiality. P.S. = Preschool. P.K. = Prekindergarten. ASL = age of initial acquisition in American Sign Language. Adop. = adopted. Vocab. = scores from picture vocabulary assessment (See Appendix A, Figure A1). VCSL = language age from Visual Communication Sign Language assessment. M = male. F = female. Disab. = disability. P.H.S = Parental hearing status. D = deaf and H = hearing. H.L. = home language. Span. = Spanish. Eng. = English. Sign. Lang. = foreign signed languages. * Lacey did not meet WWC standards but demonstrated atypical language patterns.

**Table 2 children-07-00256-t002:** Rhyming Version of Animals Crossing.

Rhyming Version
(1) SPOT—ONE—MOUSE—CROSSING (1-handshape rhyme) (2) SEE—TWO—RACCOONS—CROSSING (2-handshape rhyme) (3) JAW DROP—THREE—ROOSTERS—CROSSING (3-handshape rhyme) (4) HAIR STAND—FOUR—ZEBRAS—CROSSING (4-handshape rhyme) (5) SHOCK—FIVE—DEER—CROSSING (5-handshape rhyme) (6) WALK—FINISH! (5-handshape rhyme)

**Table 3 children-07-00256-t003:** Non-Rhyming Version of Animals Crossing.

Non-Rhyming Version
(1) SHOCK—ONE—ROOSTER—CROSSING (no handshape rhyme) (2) JAW DROP—TWO—ZEBRA—CROSSING (no handshape rhyme) (3) SPOT—THREE—DEER—CROSSING (no handshape rhyme) (4) SEE—FOUR—MICE—CROSSING (no handshape rhyme) (5) HAIR STAND—FIVE—RACCOONS—CROSSING (no handshape rhyme) (6) WALK—FINISH!

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
