# Peer review of "Effects of ASL Rhyme and Rhythm on Deaf Children’s Engagement Behavior and Accuracy in Recitation: Evidence from a Single Case Design"

_children, 2020, doi:10.3390/children7120256_

Round 1
Reviewer 1 Report
Thank you for the opportunity to review this manuscript. A study examining an intervention of ASL phonological development is greatly needed in the field of deaf education, and I was eager to review it.
Introduction section
The manuscript was well-designed throughout and provided a thorough review of the benefits of phonological with hearing and deaf children and hearing children with additional processing challenges.
The only thing that I would suggest the authors add is to update their information on p.5 line 245 , which stated that the Di Perri, 2004 article was the *first* to examine deaf children’s awareness of phonological features of signs. There were multiple other studies that had examined awareness of phonological features before this study was published in 2004, such as Bonvillian & Siedlecki, 1996; 2000, Boyes-Braem, 1990, and Conlin et al., 2000). Adding those references would strengthen this literature review.
Petitto et al. (2016) also made an essential contribution in their work, examining Visual Sign Phonology. An important point of their study is that the brain is looking out for phonological units and linguistic patterning in its environment, and it is biologically open to accepting this signal in both sound and visual units. An important part of this argument that could be stressed more in their review is focused on patterning behavior that the brain is naturally seeking. It seems to be an important piece in their argument later in discussion with how the participants seemed better able to recall vocabulary and imitate more readily, even in the deaf children who were delayed in their language.
Methods Section
A brief description of the materials used to assess the baseline of the students would be beneficial, especially with the vocabulary and the VCSL. I did not have access to “Appendix X” to not assess what the vocabulary test was. I know of the VCSL, but I do not know what’s included in the VCSL assessment. A brief summary would be helpful for other readers who know little about it and me. This would also be important since the author's discussion the VCSL as an assessment tool later in the discussion.
It would also be helpful to identify which student had the Sign Language Impairment. I presume it might be “Yair” because he was a native signer but yet had a lower vocabulary than the other two signers with very early exposure.
Results section
I thought the results were appropriate for the data. One challenge that I had was more related to the presentation of the figures.
It might help to label the figures more clearly in Figure 2. It took me a few minutes to figure out what each represented. While it was possible to figure out from the captioning which figure was which, identifying it in some way in the figure itself would add clarity.
I also found the font sizes were too small for the legends in all the figures, making them all very difficult to read. It would help older readers like myself to make the font sizes a little bigger.
How many typical children were there? In Figure 3, it states, Delayed (n=2) and Typical (n=3). In figure 5, it states Delayed (n=3) and Typical (n=2). It would help to be consistent on both or explain why there is a difference.
Discussion section
It was not clear which of the three deaf children with typical exposure to language had the Sign Language Impairment. The data presentation showed that all three showed an upward trajectory in their development of sign phonology. What was written in the discussion did not seem to match what was displayed in the figures. (p. 17, lines 581-584), some clarification in this section would be helpful.
I liked the point about sign phonology being used as a potential means to diagnose deaf children with SLI. It was a great point.
In the Engagement and Imitation section, I also really liked the point that the deaf children with delayed language had almost no imitations with the non-rhyming condition but has more imitations (about 25%) during the rhyming condition. There was also a drastic increase in deaf children with typical language ability. This section might help make the connection between Petitto et al (2016) claim that the brain is naturally seeking out patterning opportunities, and this study also provides some support for the visually based phonological patterning Petitto and colleagues cite in their data.
Limitations and Future Directions
I believe the authors make a good point about the limited validity and reliability of the VCSL, but expanding a little more about what’s included in the VCSL’s assessment would make the argument more salient. The VCSL may be limited, it’s only one of the few tools that are available to assess the signing development of young children, so the tool is not definitive. Still, it does provide us with some limited information that has value.
Summary of Review
In summary, I enjoyed reading the manuscript and thought that it was very well done. There are a few areas where I believe there could be more clarity that would strengthen the paper's overall quality. The study contributes some very important insights into the potential contributions of sign language phonology with early language development and builds off of previous studies in this area. It also stresses a critical point of using ASL phonology as another form of data to help diagnosis deaf children sign Sign Language Impairment. These areas are very underdeveloped in deaf education, and this study makes an essential contribution to furthering our knowledge base.
Reviewer 2 Report
This is an important and well-written paper. My only recommendation is that the fact that the instrument videos were researcher-produced be added to the Limitations section. Although I'm sure the researcher intended to make both sets of videos equally engaging, it is possible that the ASL-rhyme videos were signed in a more engaging manner than the non-rhyming videos given the researcher's investment in the topic. In the future, a better design would be asking a signer unfamiliar with the purposes of study to produce the videos.
